# PARTIAL TRANSPORTABILITY FOR DOMAIN GENERALIZATION

## ABSTRACT

Learning prediction models that generalize to related domains is one of the most fundamental challenges in artificial intelligence. There exists a growing literature that argues for learning invariant associations using data from multiple source domains. However, whether invariant predictors generalize to a given target domain depends crucially on the assumed structural changes between domains. Using the perspective of transportability theory, we show that invariance learning, and the settings in which invariant predictors are optimal in terms of worst-case losses, is a special case of a more general *partial* transportability task. Specifically, the partial transportability task seeks to identify / bound a conditional expectation $\mathbb{E}_{P*}[Y \mid \mathbf{x}]$ in an unseen domain $\pi^*$ using knowledge of qualitative changes across domains in the form of causal graphs and data from source domains $\pi^1, \ldots, \pi^k$. We show that solutions to this problem have a much wider generalization guarantee that subsumes those of invariance learning and other robust optimization methods that are inspired by causality. For computations in practice, we develop an algorithm that provably provides tight bounds asymptotically in the number of data samples from source domains for any partial transportability problem with discrete observables and illustrate its use on synthetic datasets.

## 1 INTRODUCTION

Generalization guarantees are central to the design of reliable machine learning models as the predictions and conclusions obtained in one or several source domains $\pi^1, \ldots, \pi^k$ (e.g. in controlled laboratory circumstances, from a specific study or population, etc.) are transported and applied elsewhere, in a domain $\pi^*$ that may differ in several aspects from that of source domains. It is apparent that what structure and what assumptions are imposed on the relationship between domains determines whether a model will generalize as intended. For example, if the target environment is arbitrary, or substantially different from the study environment, transporting predictions is difficult or even impossible.

A structural account of causation provides suitable semantics for reasoning about the structural invariances across different domains, and has been studied under the umbrella of transportability theory (Pearl & Bareinboim, 2011; Bareinboim et al., 2013; Bareinboim & Pearl, 2016). Each domain $\pi^i$ is associated with a different structural causal model (SCM) $M^i$ that differs in one or more of its component parts with respect to other domains and defines different distributions over the observed variables. In practice, the SCMs are usually not fully observable, which leads to the transportability challenge of using data from one (or more) SCMs to make inference about distributions from another SCM. A query, e.g. $\mathbb{E}_{P*}[Y \mid \mathbf{x}]$, is said to be point identified if it can be uniquely computed given available data (from one or more domains) and qualitative knowledge about the causal changes between domains in the form of selection diagrams. However, in problems of transportability, especially when no data in the target domain can be collected, the combination of qualitative assumptions and data often does not permit one to uniquely determine a given query, which is said to be non-identifiable. In such cases, partial identification methods deal with bounding a given query e.g. $l < \mathbb{E}_{P*}[Y \mid \mathbf{x}] < u$ in non-identifiable problems and may still serve an informative purpose for decision-making if $0 < l < u < 1$. Both settings have been studied in the literature. In particular, there exists an extensive set of graphical conditions and algorithms for the identifiability of observational, interventional, and counterfactuals distributions across domains from a combination of

datasets in various settings (Pearl & Bareinboim, 2011; Bareinboim et al., 2013; Bareinboim & Pearl, 2014; 2016; Lee et al., 2020; Correa & Bareinboim, 2019). For example, Lee et al. (2020) investigate the transportability of conditional causal effects, while Correa & Bareinboim (2020) investigate the transportability of soft interventions or policies, from an arbitrary combination of datasets collected under different conditions. Several methods exist also for partial identification of causal effects and counterfactuals (Balke & Pearl, 1997; Chickering & Pearl, 1996; Zhang et al., 2021) that aim at bounding insead of point-identifying a particular causal effect. Despite the generality of these results, there is still no treatment or algorithms for the partial identification of transportability queries.

In the machine learning literature, notably, a version of the transportability task is also widely studied as the problem of *domain generalization* (Wang et al., 2022). The objective is to learn a prediction function with a minimum performance guarantee on any distribution in some uncertainty set that includes potential test / target distributions (Ben-Tal et al., 2009; Gulrajani & Lopez-Paz, 2020). This problem has implicit connections to causality and SCMs if uncertainty sets of distributions are defined on the basis of "invariant correlations", such as stable conditional expectations $\mathbb{E}_{P^1}[Y \mid \mathbf{x}] = \cdots = \mathbb{E}_{P^k}[Y \mid \mathbf{x}]$ across training domains $\pi^1, \ldots, \pi^k$, to be used for prediction in a target domain $\pi^*$ and that may be learned from data sampled across sufficiently many different domains with statistical tests (Peters et al., 2016; Subbaswamy et al., 2019; Subbaswamy & Saria, 2020) or custom loss functions (Magliacane et al., 2018; Arjovsky et al., 2019; Rojas-Carulla et al., 2018; Bellot & van der Schaar, 2020). For instance, Arjovsky et al. (2019) argue for learning representations that define an invariant optimal classifier across several training datasets. Subbaswamy et al. (2019); Subbaswamy & Saria (2020) use causal graphs and identifiable interventional distributions to define invariant prediction rules across domains. Notwithstanding their wide applicability, there is little theoretical understanding of the extrapolation guarantees that can be expected from invariant prediction rules given a finite set of domains. Correlations invariant across source domains need not be invariant in a target domain; and performance guarantees, in general, depend on the structural invariances assumed for their respective SCMs.

In this paper, we start by describing the conditions under which invariant prediction rules can be expected to perform well in an arbitrary target domain from first principles using the semantics of structural causal models (Pearl, 2009; Pearl & Bareinboim, 2011). We then introduce a broader optimization problem – the task of partial transportability – whose objective is to bound, instead of point estimate, a query in an arbitrary target domain of interest, such as $\mathbb{E}_{P*}[Y \mid \mathbf{x}]$, given data from one or more source domains and qualitative knowledge about the causal changes between domains in the form of selection diagrams. We demonstrate that solutions to this problem subsume various instantiations of invariant predictors (in the conditions where these are adequate) and have a wider distributional robustness guarantee to any distribution in the target domain that is compatible with the assumed selection diagrams. For computations in practice, we show that the partial transportability task can be solved approximately for systems of variables with finite domains with a Markov Chain Monte Carlo sampling approach. The resulting bounds are sound and tight, and provide the most informative inference on a target query given the available information.

## 1.1 PRELIMINARIES

We introduce in this section some basic notations and definitions that will be used throughout the paper. We use capital letters to denote variables ($X$), small letters for their values ($x$), bold letters for sets of variables ($\mathbf{X}$) and their values ($\mathbf{x}$), and use $\Omega$ to denote their domains of definition ($x \in \Omega_X$). A conditional independence statement in distribution $P$ is written as $(\mathbf{X} \perp\!\!\!\perp \mathbf{Y} \mid \mathbf{Z})_P$. A $d$-separation statement in some graph $\mathcal{G}$ is written as $(\mathbf{X} \perp\!\!\!\perp \mathbf{Y} \mid \mathbf{Z})_\mathcal{G}$. For convenience, we denote by $P(\mathbf{x})$ probabilities $P(\mathbf{X} = \mathbf{x})$, and $\mathbb{1}\{\cdot\}$ for the indicator function equal to 1 if the statement in $\{\cdot\}$ evaluates to true, and equal to 0 otherwise. All proofs are given in the Appendix.

We use the language of structural causal models (SCMs) (Definition 7.1.1 (Pearl, 2009)) to define the semantics of causality. An SCM $M$ is a tuple $M = \langle \mathbf{V}, \mathbf{U}, \mathcal{F}, P \rangle$ where $\mathbf{V}$ is a set of endogenous variables and $\mathbf{U}$ is a set of exogenous variables. Each exogenous variable $U \in \mathbf{U}$ is distributed according to a probability measure $P(u)$. $\mathcal{F}$ is a set of functions where each $f_V \in \mathcal{F}$ determines the deterministic dependencies of $V$ on other parts of the system. That is, $v := f_V(\boldsymbol{pa}_V, \boldsymbol{u}_V)$, with $\mathbf{Pa}_V \subset \mathbf{V}$, and $\mathbf{U}_V \subset \mathbf{U}$, the exogenous sources of variation that influence $V$. With this construction, we define the potential response $\mathbf{V}(\boldsymbol{u})$ to be the solution of $\mathbf{V}$ in the model $M$ given $\mathbf{U} = \boldsymbol{u}$. Moreover, drawing values of exogenous variables $\mathbf{U}$ following the probability measure $P$

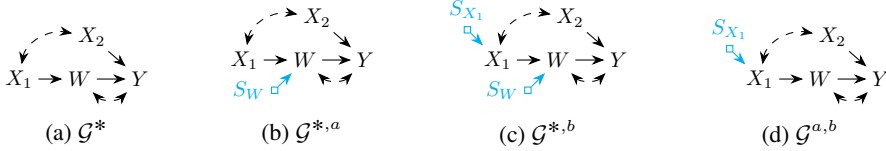

(a) $\mathcal{G}^*$     (b) $\mathcal{G}^{*,a}$     (c) $\mathcal{G}^{*,b}$     (d) $\mathcal{G}^{a,b}$

Figure 1: Example of graphs: (a) Causal graph of target domain $\pi^*$, (b) selection diagram that compares domains $\pi^*$ with $\pi^a$, (c) selection diagram that compares domains $\pi^*$ with $\pi^b$, (d) selection diagram that compares domains $\pi^a$ with $\pi^b$.

induces a joint distribution over observables given by,

$$P(\mathbf{v}) = \int_{\Omega_{\mathbf{U}}} \prod_{V \in \mathbf{V}} \mathbb{1}\{f_V(\boldsymbol{pa}_V, \boldsymbol{u}_V) = v\}dP(\boldsymbol{u}). \tag{1}$$

An SCM induces a causal graph $\mathcal{G}$ in which each variable in $\mathbf{V}$ is associated with a node; we draw a directed edge between two variables $X \to Y$ if $X \in \mathbf{Pa}_Y$ appears as an argument of $f_Y$ in the SCM, and a bidirected arrow $X \leftrightarrow Y$ if $\mathbf{U}_X \cap \mathbf{U}_Y \neq \varnothing$, that is $X$ and $Y$ share an unobserved confounder. The set of parent nodes of $\mathbf{X}$ in $\mathcal{G}$ is denoted by $pa(\mathbf{X})_{\mathcal{G}} = \bigcup_{X \in \mathbf{X}} pa(X)_{\mathcal{G}}$. Its capitalized version $Pa$ includes the argument as well, e.g. $Pa(\mathbf{X})_{\mathcal{G}} = pa(\mathbf{X})_{\mathcal{G}} \cup \mathbf{X}$. Similar definitions are used for children $ch$, descendants $de$, etc.

## 2    DOMAIN GENERALIZATION THROUGH THE LENS OF TRANSPORTABILITY

We adopt the setting of domain generalization. We assume access to $k$ source domains $\pi^1, \pi^2, \ldots, \pi^k$ with associated data distributions over a common set of variables $\mathbf{V}$ denoted $P^1(\mathbf{v}), P^2(\mathbf{v}), \ldots, P^k(\mathbf{v})$. Our focus is on a query, such as $\mathbb{E}_{P*}[Y \mid \mathbf{x}]$, to be evaluated in a target domain $\pi^*$ (potentially) different from source domains, where typically $Y$ is an outcome variable, $\mathbf{X}$ is a set of covariates, and $Y \cup \mathbf{X} = \mathbf{V}$.

For concreteness, consider a medical study where patient data was collected under different treatment protocols in an attempt to assess, in a target hospital $\pi^*$, the prognosis of neurodegenerative diseases such as Alzheimer's in patients with a number of existing conditions. In the causal graph in Fig. 1a, $X_1$ and $X_2$ are treatments for hypertension and clinical depression, respectively, both known to be causes of neurodegenerative diseases $Y$. In the case of hypertension, the effect is mediated by blood pressure $W$, whose effect on neurodegenerative diseases is confounded, since both conditions share important confounding factors such as physical activity levels and diet patterns (Skoog & Gustafson, 2006) (graphically encoded through the bidirected arrows). Hypertension and clinical depression are not known to affect each other (no direct link between them), although it's common for patients with clinical depression to simultaneously be at risk of hypertension (Meng et al., 2012). In this example, we have access to an observational study conducted in a hospital $\pi^a$, and to hospital $\pi^b$ following different guidelines for the administration of $X_1$, both of which however are known to have a different high blood pressure $W$ incidence than that in $\pi^*$. These differences are called domain discrepancies (Pearl & Bareinboim, 2011).

**Definition 1** (Domain Discrepancy, (Pearl & Bareinboim, 2011))**.** *Let $\pi_a$ and $\pi_b$ be domains associated, respectively, with SCMs $M^a$ and $M^b$ and causal diagrams $\mathcal{G}^a$ and $\mathcal{G}^b$. We denote by $\Delta^{a,b} \subset \mathbf{V}$ a set of variables such that, for every $V_i \in \Delta^{a,b}$, there might exist a discrepancy if $f_{V_i}^a \neq f_{V_i}^b$ or $P^a(\mathbf{U}_i) \neq P^b(\mathbf{U}_i)$.*

**Definition 2** (Selection diagram, (Pearl & Bareinboim, 2011))**.** *Given domain discrepancies $\Delta^{a,b}$ between two domains $\pi^a$ and $\pi^b$ and a causal graph $\mathcal{G}^a = (\mathbf{V}, \mathbf{E})$, let $\mathbf{S} = \{S_V : V \in \Delta^{a,b}\}$ be called selection nodes. Then, a selection diagram $\mathcal{G}^{a,b}$ is defined as a graph $(\mathbf{V} \cup \mathbf{S}, \mathbf{E} \cup \{S_V \to V\}_{S_V \in \mathbf{S}})$.*

Selection nodes locate the mechanisms where structural discrepancies between the two domains are suspected to take place. The absence of a selection node pointing to a variable represents the assumption that the mechanism responsible for assigning value to that variable is identical in both domains. In the medical example above, Fig. 1b shows a selection diagram comparing domains $\pi^a$ and $\pi^*$ in which the $S_W$ node indicates a structural difference in the assignment of $W$, either

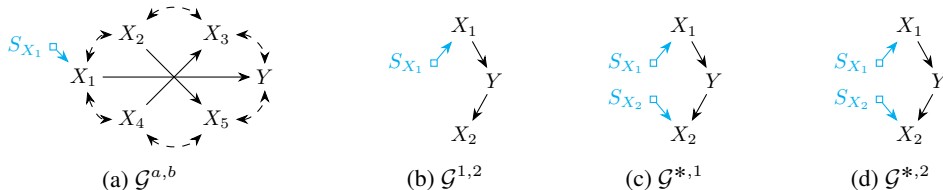

(a) $\mathcal{G}^{a,b}$        (b) $\mathcal{G}^{1,2}$        (c) $\mathcal{G}^{*,1}$        (d) $\mathcal{G}^{*,2}$

Figure 2: Graphs used in Sec. 2.1.

$f_W^* \neq f_W^a$ and / or $P^*(u_W) \neq P^a(u_W)$, but not in the assignment of other variables, for instance $f_Y^* = f_Y^a$ and $P^*(u_Y) = P^a(u_Y)$. Fig. 1c and Fig. 1d are selection diagrams that compare domains $(\pi^b, \pi^*)$ and $(\pi^a, \pi^b)$ respectively.

## 2.1 INVARIANCE LEARNING FOR DOMAIN GENERALIZATION

It is apparent that there is a degree of unidentifiability in optimal prediction rules in a target domain depending on the structural differences between it and the available data. A natural objective for a chosen prediction function is to minimize worst-case losses over an uncertainty set of potential target distributions that are compatible with a set of selection diagrams $\{\mathcal{G}^{i,*} : i = 1, \ldots, k\}$

$$\arg\min_f \max_{M \in \mathcal{M}(\mathcal{G}*)} \mathbb{E}_{P^M}[(Y - f(\mathbf{X}))^2], \tag{2}$$

where $\mathcal{M}(\mathcal{G}*)$ is the family of SCMs compatible with the causal graph $\mathcal{G}*$. In the literature on domain generalization, selection diagrams $\{\mathcal{G}^{i,*} : i = 1, \ldots, k\}$ are mostly implicit, and it is common to define predictors agnostic of assumptions on the underlying causal structure of the target domain, and instead exploit invariances with respect to source domains, see e.g. the proposals of (Arjovsky et al., 2019; Peters et al., 2016; Lu et al., 2021; Rojas-Carulla et al., 2018; Magliacane et al., 2018). This section studies the generalization guarantees of a common class of invariant predictors in the language of selection diagrams.

**Definition 3** (Invariant predictor). Given selection diagrams $\{\mathcal{G}^{i,j} : i, j = 1, \ldots, k\}$, an invariant predictor is given by $\mathbb{E}_P[Y \mid \mathbf{z}]$ where $(Y \perp\!\!\!\perp \mathbf{S} \mid \mathbf{Z})_{\mathcal{G}^{i,j}}$ for $i, j = 1, \ldots, k$ and the expectation is taken with respect to any $P$ among source domain distributions.[1]

Invariant predictors define stable conditional expectations, i.e. $\mathbb{E}_{P^1}[Y \mid \mathbf{z}] = \cdots = \mathbb{E}_{P^k}[Y \mid \mathbf{z}]$. We use the notion of domain-independent Markov blankets to define *optimal* invariant predictors.

**Definition 4** (Domain-independent Markov blankets). Given a set of selection diagrams $\{\mathcal{G}^{i,j} : i, j = 1, \ldots, k\}$, the set of domain-independent Markov blankets for $Y \in \mathbf{V}$ is given by the set of $\mathbf{Z} \subset \mathbf{V}$ such that (1) $(Y \perp\!\!\!\perp \mathbf{S} \mid \mathbf{Z})_{\mathcal{G}^{i,j}}$ for $i, j = 1, \ldots, k$ and (2) $(W \perp\!\!\!\perp Y \mid \mathbf{Z}\backslash W)_{\mathcal{G}^{i,j}}$ for $i, j = 1, \ldots, k$ and all $W \in \mathbf{Z}$.

Domain-independent Markov blankets are designed to be minimal, in the sense that no proper subset of them satisfies conditions (1) and (2), and informative for predicting $Y$ while defining stable conditional distributions across source domains. In general, such a set is not guaranteed to exist. For example, in Fig. 1b there is no set (and by implication no invariant predictor) that separates $Y$ from all selection nodes, i.e. condition (1) in Def. 4 is violated for any subset of $\mathbf{V}$. Moreover, contrary to the conventional Markov blanket (Pearl & Paz, 1985), it is not guaranteed to be unique. For example, in Fig. 2a both $\{X_1, X_2, X_5\}$ and $\{X_1, X_3, X_4\}$ are domain invariant Markov blankets. Which one is most informative to predict $Y$ is undecidable from the graph structure alone, i.e. it depends the exact functional associations between variables.

**Proposition 1** (Optimal invariant predictor). *Given selection diagrams $\{\mathcal{G}^{i,j} : i, j = 1, \ldots, k\}$, the optimal invariant predictor is defined as the minimizer of $\mathbb{E}_{P^i}[(Y - f(\mathbf{Z}))^2]$ across all $i = 1, \ldots, k$,*

---

[1]Other invariance assumptions have also been made, e.g. $(\mathbf{Z} \perp\!\!\!\perp \mathbf{S})_{\mathcal{G}^{i,j}}$ and $(Y \perp\!\!\!\perp \mathbf{S} \mid \mathbf{X})_{\mathcal{G}^{i,j}}$ for problems where only the distribution of some covariates is expected to change across domains (Muandet et al., 2013), or such as $(\mathbf{Z} \perp\!\!\!\perp \mathbf{S} \mid Y)_{\mathcal{G}^{i,j}}$ (Li et al., 2018). Problems where the magnitude of changes is assumed to be bounded, i.e. $\mid P^*(y \mid \mathbf{x}) - P^i(y \mid \mathbf{x}) \mid \leqslant c$, instead of restricting the $d$-separation statements involving $\mathbf{S}$ have been studied by (Rothenhäusler et al., 2021).

*and belongs to the set of invariant predictors for which $\mathbf{Z}$ is a domain-independent Markov blanket for $Y \in \mathbf{V}$.*

Invariant predictors may be desirable due to their stability, although, it is apparent that the extent to which predictors will generalize outside of source domains depends on the structure of $\mathcal{M}(\mathcal{G}^*)$ and, in particular, differences in structure with respect to source domains. In general, structural invariances across source domains need not hold outside of source domains. For example, given two source domains $\pi^1, \pi^2$ described by $\mathcal{G}^{1,2}$ in Fig. 2b, it holds that $\mathbb{E}_{P^1}[Y \mid x_1, x_2] = \mathbb{E}_{P^2}[Y \mid x_1, x_2]$ is the invariant predictor, which may not be optimal in a target domain $\pi^*$ if the same invariance doesn't hold. For example, for the selection diagrams $\mathcal{G}^{*,1}$ and $\mathcal{G}^{*,2}$ given in Fig. 2c and Fig. 2d $\mathbb{E}_{P^1}[Y \mid x_1, x_2] \neq \mathbb{E}_{P*}[Y \mid x_1, x_2]$.

In fact, the generalization error of the optimal invariant predictor, here denoted $\mathbb{E}_{P^1}[Y \mid \mathbf{z}]$, can be written as

$$
\begin{aligned}
\max_{M \in \mathcal{M}(\mathcal{G}*)} & \mathbb{E}_{PM}\left[(Y - \mathbb{E}_{P^1}[Y \mid \mathbf{Z}])^2\right] = \\
\max_{M \in \mathcal{M}(\mathcal{G}*)} & \left(\mathbb{E}_{PM}\left[(Y - \mathbb{E}_{PM}[Y \mid \mathbf{X}])^2\right] + \mathbb{E}_{PM}\left[(\mathbb{E}_{PM}[Y \mid \mathbf{X}] - \mathbb{E}_{P^1}[Y \mid \mathbf{Z}])^2\right]\right),
\end{aligned} \quad (3)
$$

where the first term on the RHS is the expected conditional variance and is in general irreducible, and the second term on the RHS quantifies the difference between the invariant predictor and the optimal prediction rule. This second term may be arbitrarily large for a general class of SCMs $\mathcal{M}(\mathcal{G}^*)$ with arbitrary differences with source domains. As a consequence, optimality of invariant predictors as solutions to Eq. (2) is limited in general to specific scenarios.

**Proposition 2** (Generalization guarantees of optimal invariant predictors). *Given a set of selection diagrams $\{\mathcal{G}^{i,j} : i, j = 1, \ldots, k\}$, let $\Delta = \bigcup_{i,j} \Delta^{i,j}$ be the set of variables in $\mathbf{V}$ whose causal mechanisms differ between any two source domains, and let $\mathbf{S} = \{S_V : V \in \Delta\}$. Consider the robust optimization problem in Eq. (2). The optimal invariant predictor is a solution if selection nodes in all selection diagrams $\{\mathcal{G}^{i,*} : i = 1, \ldots, k\}$ are given by $\mathbf{S}$ with edges $\{S_V \to V\}_{S_V \in \mathbf{S}}$.*

In words, an optimal invariant predictor has lowest generalization error in the sense of Eq. (2) only in the space of target SCMs $\mathcal{M}(\mathcal{G}^*)$ with the *same* structural invariances observed across source domains. Otherwise, in general better predictors are achievable. This observation includes predictors using causal parents as a conditioning set (often understood as desirable for domain generalization) which, similarly, define robust predictors for a target domain if invariance in the association between causal parents and outcomes is assumed. For example, in Fig. 1, $\mathbb{E}_{P^a}[Y \mid \boldsymbol{pa}_Y] \neq \mathbb{E}_{P^b}[Y \mid \boldsymbol{pa}_Y]$, $\mathbb{E}_{P^a}[Y \mid \boldsymbol{pa}_Y] \neq \mathbb{E}_{P*}[Y \mid \boldsymbol{pa}_Y]$, and $\mathbb{E}_{P^b}[Y \mid \boldsymbol{pa}_Y] \neq \mathbb{E}_{P*}[Y \mid \boldsymbol{pa}_Y]$, and thus predictors based on causal parents may not be robust or optimal, in general. In particular, a prediction function of the form $\mathbb{E}_{P^1}[Y \mid \boldsymbol{pa}_Y]$ is a solution to the robust optimization problem in Eq. (2) if and only if it is the optimal invariant predictor and $\{\mathcal{G}^{i,*} : i = 1, \ldots, k\}$ is defined as in Prop. 2.

Moreover, independently of whether solutions to a worst-case optimization problem can be found, they say nothing about the *range* of values optimal prediction functions $\mathbb{E}_{P*}[Y \mid \mathbf{x}]$ may take in other distributions $P^*$ away from the worst-case. In the following section, we attempt to define predictors and ranges of predictors with guarantees to arbitrary sets $\mathcal{M}(\mathcal{G}^*)$.

## 3 PARTIAL TRANSPORTABILITY OF STATISTICAL RELATIONS

The uncertainty and inherent under-identifiability of solutions to domain generalization problems motivates us to define the task of *partial* transportability, that extends the literature on domain generalization by considering bounds on the value of arbitrary queries $\mathbb{E}_{P*}[Y \mid \mathbf{x}]$ in arbitrary target domains $\pi^*$ defined by a set of selection diagrams $\{\mathcal{G}^{i,*} : i = 1, \ldots, k\}$.

**Task** (Partial Transportability). *Derive a tight bound $[l, u]$ over a query of the form $\mathbb{E}_{P*}[Y \mid \mathbf{x}]$ with knowledge of selection diagrams $\{\mathcal{G}^{*,i} : i = 1, \ldots, k\}$, a corresponding collection of data distributions $\{P^i(\mathbf{v}) : i = 1, \ldots, k\}$, and set of intervals $\{I_j : V_j \in \bigcup_i \Delta^{*,i}\}$ that define potential constraints on probabilities in the target domain. Algorithmically, this may be written as a solution*

*to the following optimization problem,*

$$\min_{M \in \mathcal{M}(\mathcal{G}^*)} / \max \mathbb{E}_{PM}[Y \mid \mathbf{x}], \quad \text{such that} \quad \forall V \notin \Delta^{*,i} : f_V^* = f_V^i, P^*(\boldsymbol{u}_V) = P^i(\boldsymbol{u}_V),$$

$$\text{and} \quad \forall V \in \bigcup_i \Delta^{*,i}, P^*(v \mid \boldsymbol{pa}_V) \in I_V. \tag{4}$$

In words, the task is to evaluate the minimum and maximum values over all possible SCMs $M$ compatible with $\{\mathcal{G}^{*,i} : i = 1, \ldots, k\}$ that define the structurally invariant mechanisms in the system and (potentially uninformative, i.e., $I_V \in [0, 1]$) assumptions about target-specific probabilities.

For example, given the causal description of the protocols presented in the introductory medical example and Fig. 1, the question might be how to combine these various datasets to predict an individual's risk of developing neurodegenerative diseases in $\pi^*$? The optimal prediction function is given by $\mathbb{E}_{P*}[Y \mid w, x_1, x_2]$ (under mean squared error losses), which may be written as,

$$\mathbb{E}_{P*}[Y \mid w, x_1, x_2] = \sum_{y \in \Omega_Y} yP^*(y, w, x_1, x_2) / \sum_{y \in \Omega_Y} P^*(y, w, x_1, x_2), \tag{5}$$

where $P^*(y, w, x_1, x_2)$ is equal to,

$$\int_{\Omega_{\mathbf{U}}} \underbrace{\mathbb{1}\{f_Y^*(w, x_2, u_{wy}) = y\}}_{\text{matches RCT } \pi^b} \underbrace{\mathbb{1}\{f_W^*(x_1, u_{wy}) = w\}}_{\text{specific to } \pi^*} \underbrace{\mathbb{1}\{f_{X_1, X_2}^*(u_{x_1 x_2}) = x_1, x_2\}}_{\text{matches hospital } \pi^a} dP^*(\boldsymbol{u}). \tag{6}$$

This is a mixture of terms for which data from source domains can be leveraged, for example $f_Y^*(w, x_2, u_{wy}) = f_Y^a(w, x_2, u_{wy})$ (superscripts denote domain), but also involves unobserved confounders $u_{wy}$ and $u_{x_1 x_2}$ that cannot be marginalized out, and terms that are specific to $\pi^*$. In addition, although $P^*(w \mid x_1)$ is known to differ in our target medical study it may be the case that we have some domain knowledge that constrains it, e.g. $P^*(w \mid x_1) \in [0.2, 0.7] =: I_w$ (if left undetermined, $I_w := [0, 1]$), and can be used to further inform a target query.

The following proposition show that the solution of the partial transportability task defines an interval that contains the invariant predictor and, by definition, also the optimal "worst-case" predictor across $\mathcal{M}(\mathcal{G}^*)$.

**Proposition 3.** *For a given set of selection diagrams, let $[l(\mathbf{x}), u(\mathbf{x})]$ denote the solution of the partial transportability task for the query $\mathbb{E}_{PM}[Y \mid \mathbf{x}], M \in \mathcal{M}(\mathcal{G}^*)$ and $\mathbb{E}_{P^1}[Y \mid \mathbf{z}], \mathbf{Z} \subseteq \mathbf{X}$ be the invariant predictor. Then, $\mathbb{E}_{P^1}[Y \mid \mathbf{z}] \in [l(\mathbf{x}), u(\mathbf{x})]$. Moreover, by definition $\mathbb{E}_{PM}[Y \mid \mathbf{x}] \in [l(\mathbf{x}), u(\mathbf{x})]$ for a particular "worst-case" member $M \in \mathcal{M}(\mathcal{G}^*)$.*

In general, there is no reason to believe that the invariant predictor has any special performance guarantee among other solutions in $[l(\mathbf{x}), u(\mathbf{x})]$. For example, the worst-case loss in Eq. (3) is not, in general, smallest when $\mathbb{E}_{PM}[Y \mid \mathbf{x}] \neq \mathbb{E}_{P^1}[Y \mid \mathbf{z}]$. An alternative is to exploit the solutions to the partial transportability task to define the median of $[l(\mathbf{x}), u(\mathbf{x})]$ as a general predictor for domain generalization problems.

**Proposition 4.** *For a given set of selection diagrams and data, let $[l(\mathbf{x}), u(\mathbf{x})]$ denote the solution of the partial transportability task for the query $\mathbb{E}_{PM}[Y \mid \mathbf{x}], M \in \mathcal{M}(\mathcal{G}^*)$. Then,*

$$\max_{M \in \mathcal{M}(\mathcal{G}^*)} \mathbb{E}_{PM}\Big[\Big(Y - \operatorname*{med}_{M \in \mathcal{M}(\mathcal{G}^*)} \mathbb{E}_{PM}[Y \mid \mathbf{X}]\Big)^2\Big]$$

$$\leqslant \max_{M \in \mathcal{M}(\mathcal{G}^*)} \left( \mathbb{E}_{PM}[(Y - \mathbb{E}_{PM}[Y \mid \mathbf{X}])^2] + \frac{1}{4}\mathbb{E}_{PM}[(u(\mathbf{X}) - l(\mathbf{X}))^2] \right).$$

*Under the condition that the irreducible error $\mathbb{E}_{PM}[(Y - \mathbb{E}_{PM}[Y \mid \mathbf{X}])^2]$ is constant across $M \in \mathcal{M}(\mathcal{G}^*)$, $\operatorname{med}_{M \in \mathcal{M}(\mathcal{G}^*)} \mathbb{E}_{PM}[Y \mid \mathbf{X}]$ provably solves the robust optimization problem Eq. (2).*

This proposition says that the error of the median is, at most, off from the optimal predictor by "half the range of possible values of $\mathbb{E}_{PM}[Y \mid \mathbf{x}]$ compatible with the data and assumptions" and that this error is optimal in the worst case (under some assumptions on how the expected conditional variance is allowed to vary). This result is important because it applies to any set of target causal graph, source domains, and selection diagrams. Note, however, that this does not mean that the median is always superior to the optimal invariant predictor: in selected settings where the expected conditional variance changes across domains we may still have the optimal invariant predictor being a better worst-case solution.

# 4 ALGORITHMS FOR PARTIAL TRANSPORTABILITY

This section presents algorithms to solve the partial transportability task for SCMs with *discrete observables*, that is each $V \in \mathbf{V}$ taking values in a finite space of outcomes, while each $U \in \mathbf{U}$ associated with an arbitrary probability density function $P(\mathbf{u})$.

A first step in our argument will be to decompose a chosen query into smaller factors so as to infer which factors can be matched across domains and point-identified from data, and subsequently re-parameterize unmatched factors by a special family of SCMs to make the bounding problem tractable. We use the concept of $c$-components and $C$-factors developed by Tian & Pearl (2002). The set $\mathbf{V}$ can be partitioned into $c$-components such that two variables are assigned to the same set $\mathbf{C} \subset \mathbf{V}$ if and only if they are connected by a bi-directed path in $\mathcal{G}$. In addition let $\mathbf{U_C} = \bigcup_{V_i \in \mathbf{C}} \mathbf{U}_i$ denote the set of exogenous variables that are parents of any $V \in \mathbf{C}$. For example, the graph in Fig. 1a induces $c$-components $\{X_1, X_2\}$ and $\{W, Y\}$.

For any set $\mathbf{C} \subseteq \mathbf{V}$, let $Q^i[\mathbf{C}](pa(\mathbf{C}))$ denote the $C$-factor of $\mathbf{C}$ in domain $\pi^i$ which is defined by,

$$Q^i[\mathbf{C}](\boldsymbol{pa_c}) = \int_{\Omega_{\mathbf{U_C}}} \prod_{V \in \mathbf{C}} \mathbb{1}\{f_V^i(\boldsymbol{pa}_V, \boldsymbol{u}_V) = v\} dP^i(\boldsymbol{u_C}). \tag{7}$$

Moreover, let $\mathbb{C}$ denote the collection of $c$-components, then $P(\mathbf{v}) = \prod_{\mathbf{C} \in \mathbb{C}} Q[\mathbf{C}]$ and $Q[\mathbf{C}] = P(\mathbf{c} \mid \boldsymbol{pa_C})$ (we omit the dependence of each $C$-factor on $pa(\mathbf{C})$ for readability). This construction is useful because the joint distribution may be factorized according to the $c$-components of $\mathcal{G}$ and its factors matched across domains (Tian & Pearl, 2002; Correa & Bareinboim, 2019).

**Lemma 1.** *Let $\mathcal{G}^{a,b}$ be a selection diagram for the SCMs $M^a$ and $M^b$, then $Q^a[\mathbf{C}] = Q^b[\mathbf{C}]$ if $\mathcal{G}^{a,b}$ does not contain selection nodes $S_V$ pointing to any variable in $V \in \mathbf{C}$.*

For example, for the selection diagram in Fig. 1b, $P^*(\mathbf{v}) = Q^*[X_1, X_2]Q^*[W, Y]$ where by Lem. 1 $Q^*[X_1, X_2] = Q^a[X_1, X_2] = P^a(x_1, x_2)$, since the is no $S$-node pointing to $X_1$ or $X_2$. In turn, $Q^*[W, Y] \neq Q^a[W, Y]$ because of the selection node pointing to $W$. Note, however, that $Q^*[W, Y]$ defined as in Eq. (7) involves terms, e.g. $\mathbb{1}\{f_Y(w, x_1, u_{wy}) = y\}$, that are invariant across domains since the absence of an $S$-node into $Y$ denotes invariance in causal mechanisms, and for which $P^a(\mathbf{v})$ may be used for estimation. We discuss next a re-parameterization of $C$-factors $Q^*[\mathbf{C}]$ that cannot be matched across domains with the goal of defining a tractable constrained optimization problem to bound $Q^*[\mathbf{C}]$.

**Proposition 5.** *Let $M$ be an arbitrary SCM with graph $\mathcal{G}$ and let $\mathbf{C}$ be any c-component. Then, there exists a corresponding SCM $N$ with finite exogenous domain compatible with $\mathcal{G}$ such that $Q^M[\mathbf{C}] = Q^N[\mathbf{C}]$, where for every exogenous variable $U \in \mathbf{U_C}$, its cardinality $d_U = |\Omega_{Pa(\mathbf{C})}|$.*

This proposition shows that SCMs with discretely-valued exogenous variables are expressive enough to represent $C$-factors $Q[\mathbf{C}]$ irrespective of the true underlying data generating mechanism. From an optimization perspective, this is useful because it allows us to consistently parameterize $C$-factors and make inference on its distribution in a well-defined latent variable model (Rosset et al., 2017; Zhang et al., 2021). As an example, consider the introductory example with $\{X_1, X_2, Y, W\}$ binary and causal graphs in Fig. 1. $Q^*[W, Y]$ defined using Eq. (7) can also be written as:

$$\sum_{u_{wy}, u_y, u_w} \mathbb{1}\{f_Y^a(w, x_2, u_{wy}, u_y) = y\} \mathbb{1}\{f_W^*(x_1, u_{wy}, u_w) = w\} P^a(u_{wy}, u_y) P^*(u_w), \tag{8}$$

where $|\Omega_{U_{wy}}| = |\Omega_{U_w}| = |\Omega_{U_y}| = |\Omega_{X_1}| \cdot |\Omega_{X_2}| \cdot |\Omega_W| \cdot |\Omega_Y| = 16$; the function $f_V$ is a mapping between finite domains $\Omega_{\mathbf{Pa}_V} \times \Omega_{\mathbf{U}_V} \mapsto \Omega_V$ for $V \in \{W, Y\}$. Moreover, we have used the structural invariances encoded by the selection diagrams in Fig. 1 to match causal mechanisms and exogenous probabilities between domains. In particular, $P^a(u_y) = P^b(u_y) = P^*(u_y)$ by definition of the selection diagrams $\mathcal{G}^{a,*}$ and $\mathcal{G}^{b,*}$. Although discretely-valued causal mechanisms and exogenous probabilities imply well-defined parameters to optimize over, the partial transportability task remains a difficult constrained optimization problem.

## 4.1 APPROXIMATIONS VIA GIBBS SAMPLING

We follow (Chickering & Pearl, 1996; Zhang et al., 2021; Bellot et al., 2022) and take a Bayesian perspective to approximating bounds $[l(\mathbf{x}), u(\mathbf{x})]$. We evaluating credible intervals $P(l(\mathbf{x}) <$

$\mathbb{E}_{P*}[Y \mid \mathbf{x}] < u(\mathbf{x}) \mid \bar{\mathbf{v}}) = 1 - \alpha$ on the posterior of $\mathbb{E}_{P*}[Y \mid \mathbf{x}]$ by approximating the expectation,

$$\mathbb{E}[\mathbb{1}\{l(\mathbf{x}) < \mathbb{E}_{P*}[Y \mid \mathbf{x}] < u(\mathbf{x})\} \mid \bar{\mathbf{v}}] = P(l(\mathbf{x}) < \mathbb{E}_{P*}[Y \mid \mathbf{x}] < u(\mathbf{x}) \mid \bar{\mathbf{v}}) \qquad (9)$$

provided with finite samples $\bar{\mathbf{v}} := (\bar{\mathbf{v}}_{\pi^1}, \ldots, \bar{\mathbf{v}}_{\pi^k})$, where $\bar{\mathbf{v}}_{\pi^i} = \{\mathbf{v}_{\pi^i}^{(j)} : j = 1, \ldots, n_i\}$ are $n_i$ independent sampled collected in domain $\pi^i$ and a set of selection diagrams $\{\mathcal{G}^{*,i} : i = 1, 2, \ldots, k\}$ using Gibbs sampling. Following the arguments in the previous section the query may be reduced to bounding a $C$-factor of the form,

$$\omega_{obj} := Q^*[\mathbf{C}] = \sum_{U \in \mathbf{U_C}} \sum_{u=1,\ldots,d_U} \prod_{V \in \mathbf{C}} \mathbb{1}\{\xi_V^{(\boldsymbol{pa}_V, \boldsymbol{u}_V)} = v\} \prod_{U \in \mathbf{U_C}} \theta_u. \qquad (10)$$

that are parameterized by $\boldsymbol{\xi} = \{\xi_V^{(\boldsymbol{pa}_V, \boldsymbol{u}_V)} : V \in \mathbf{C}, \mathbf{Pa}_V \subset \mathbf{V}, \mathbf{U}_V \subset \mathbf{U_C}\}$ and $\boldsymbol{\theta} = \{\theta_u : U \in \mathbf{U_C}\}$ that represent causal functional assignments and exogenous probabilities, respectively. We have dropped the domain indicator "*" from the definition of parameters for readability.

For every $V \in \mathbf{V}, \forall \boldsymbol{pa}_V, \mathbf{u}_V$, the functional assignment parameters $\xi_V^{(\boldsymbol{pa}_V, \boldsymbol{u}_V)}$ are drawn uniformly in the discrete domain $\Omega_V$. For every $U \in \mathbf{U}$, exogenous probabilities $\boldsymbol{\theta}_U$ with dimension $d_U = |\Omega_{Pa(\mathbf{C})}|$ are drawn from a prior Dirichlet distribution,

$$\boldsymbol{\theta}_U = (\theta_1, \ldots, \theta_{d_U}) \sim \text{Dirichlet}(\alpha_1, \ldots, \alpha_{d_U}), \qquad (11)$$

with hyperparameters $\alpha_1, \ldots, \alpha_{d_U}$. The Gibbs sampler starts with some initial value for all latent quantities $(\boldsymbol{u}, \boldsymbol{\xi}, \boldsymbol{\theta})$ in the expression of $\omega_{obj}$, and iterates over the following sampling steps, each parameter conditioned on the current values of the remaining terms in the parameter vector.

1. *Sample* $\mathbf{u}$. Let $u \in \Omega_U, U \in \mathbf{U_C}$. For each observed data example across all domains $\mathbf{v}^{(n)} \in \bar{\mathbf{v}}$, $n = 1, \ldots, \sum_i n_i$, we sample corresponding exogenous variables $U \in \mathbf{U_C}$ from the conditional distribution,

$$P(\mathbf{u}^{(n)} \mid \mathbf{v}^{(n)}, \boldsymbol{\xi}, \boldsymbol{\theta}) \propto P(\mathbf{u}^{(n)}, \mathbf{v}^{(n)} \mid \boldsymbol{\xi}, \boldsymbol{\theta}) = \prod_{V \in \mathbf{C}} \mathbb{1}\{\xi_V^{(\boldsymbol{pa}_V^{(n)}, \boldsymbol{u}_V^{(n)})} = v^{(n)}\} \prod_{U \in \mathbf{U_C}} \theta_u. \qquad (12)$$

2. *Sample* $\boldsymbol{\xi}$. Parameters $\boldsymbol{\xi}$ define deterministic causal mechanisms. For a given parameter $\xi_V^{(\boldsymbol{pa}_V, \boldsymbol{u}_V)} \in \boldsymbol{\xi}$ its conditional distribution is given by $P(\xi_V^{(\boldsymbol{pa}_V, \boldsymbol{u}_V)} = v \mid \bar{\mathbf{v}}, \bar{\mathbf{u}}) = 1$ if there exists a sample $(\mathbf{v}^{(n)}, \mathbf{pa}_V^{(n)}, \mathbf{u}^{(n)})$ for some $n$, where $n$ iterates over the samples of $\mathbf{u}$ from step 1 and $\mathbf{v}$ associated with the *subset* of domains in which exogenous probabilities match the target domain, such that $\xi_V^{(\boldsymbol{pa}_V^{(n)}, \boldsymbol{u}_V^{(n)})} = v^{(n)}$. Otherwise, $P(\xi_V^{(\boldsymbol{pa}_V, \boldsymbol{u}_V)} = v \mid \bar{\mathbf{v}}, \bar{\mathbf{u}})$ is given by a uniform discrete distribution over its domain $\Omega_V$.

3. *Sample* $\boldsymbol{\theta}$. Let $\boldsymbol{\theta}_U = (\theta_1, \ldots, \theta_{d_U}) \in \boldsymbol{\theta}$ be the parameters that define the probability vector of possible values of variables $U \in \mathbf{U_C}$. Its conditional distribution is given by,

$$\theta_1, \ldots, \theta_{d_U} \mid \bar{\mathbf{v}}, \bar{\mathbf{u}} \sim \text{Dirichlet}\left(\alpha_1 + \sum_n \mathbb{1}\{u^{(n)} = u_1\}, \ldots, \alpha_{d_U} + \sum_n \mathbb{1}\{u^{(n)} = u_{d_U}\}\right), \qquad (13)$$

where, similarly, $n$ iterates over the samples of $\mathbf{u}$ from step 1 associated with the subset of domains in which exogenous probabilities match the target domain.

In the above, we have described the conditional distributions of parameters that can be matched across domains, and therefore estimated from the subset of relevant available data. By the definition of the partial transportability task, parameters that are specific to the target domain $\pi^*$ are constrained to lie in an assumed interval, e.g. $P^*(v \mid \boldsymbol{pa}_V) = \sum_{\boldsymbol{u}_V} \mathbb{1}\{\xi_V^{(\boldsymbol{pa}_V, \boldsymbol{u}_V)} = v_i\} \prod_{U \in \mathbf{U}_V} \theta_u \in I_V \subseteq [0, 1]$, or else left unspecified. In the first case, parameters are sampled independently and uniformly in the space defined by the constraints and in the second case, they are sampled independently and uniformly in their domain of definition, i.e. $\xi_V^{(\boldsymbol{pa}_V, \boldsymbol{u}_V)} \in \Omega_V, \theta_u \in \Omega_U$, in every step of the sampler.

Iterating this procedure forms a Markov chain with the invariant distribution to be the target posterior distribution $P(\boldsymbol{u}, \boldsymbol{\xi}, \boldsymbol{\theta} \mid \bar{\mathbf{v}})$. $P(\omega_{obj} \mid \bar{\mathbf{v}})$ is then approximated by plugging the MCMC samples into Eq. (10). The upper and lower $\alpha$ quantile among $T$ samples of $P(\omega_{obj} \mid \bar{\mathbf{v}})$, when combined with the

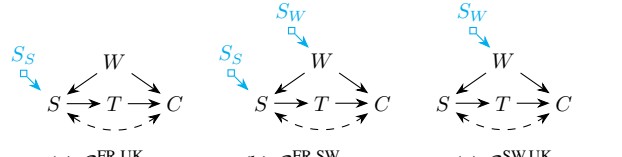

| | $\pi^{\text{FR}}$ |
|---|---|
| $\mathbb{E}_{P^{\text{UK}}}[c\mid t, w]$ | .1502 (.005) |
| $\mathbb{E}_{P^{\text{UK}}}[c\mid t, w, s]$ | .1448 (.004) |
| $\text{med}(\hat{l}_0, \hat{u}_0)$ | .1361 (.005) |

(a) $\mathcal{G}^{\text{FR,UK}}$     (b) $\mathcal{G}^{\text{FR,SW}}$     (c) $\mathcal{G}^{\text{SW,UK}}$     (d) Performance comparisons.

Figure 3: (a-c) selection diagrams that compare domain $\pi^{\text{FR}}$ with $\pi^{\text{UK}}$, $\pi^{\text{FR}}$ with $\pi^{\text{SW}}$, and $\pi^{\text{SW}}$ with $\pi^{\text{UK}}$, respectively. (d) gives mean squared error for cancer prediction on a sample of data from $P^{\text{FR}}$.

identified $C$-factors that form $\mathbb{E}_{P*}[y \mid \mathbf{x}]$, gives us a $(1-\alpha)$ credible interval $\hat{l}_\alpha < \mathbb{E}_{P*}[y \mid \mathbf{x}] < \hat{u}_\alpha$ defined by,

$$\hat{l}_\alpha(\mathbf{x}) = \sup\{x : \sum_t \mathbb{1}\{\mathbb{E}_{P*}[Y \mid \mathbf{x}]^{(t)} \leqslant x\} = \alpha/2\}, \tag{14}$$

$$\hat{u}_\alpha(\mathbf{x}) = \inf\{x : \sum_t \mathbb{1}\{\mathbb{E}_{P*}[Y \mid \mathbf{x}]^{(t)} \leqslant x\} = 1 - \alpha/2\}. \tag{15}$$

The following Theorem shows that credible intervals $[\hat{l}_0(\mathbf{x}), \hat{u}_0(\mathbf{x})]$ converge to the true bounds $[l(\mathbf{x}), u(\mathbf{x})]$ for the unknown query $\mathbb{E}_{P*}[Y \mid \mathbf{x}]$ and are, moreover, maximally informative, in the sense that we can always construct two data generating mechanisms $M^1, M^2$ for the target domain that are compatible with our current knowledge of the world such that $\mathbb{E}_{P_{M^1}}[Y \mid \mathbf{x}] = l$ and $\mathbb{E}_{P_{M^2}}[y \mid \mathbf{x}] = u$.

**Theorem 1.** *The solution $[l(\mathbf{x}), u(\mathbf{x})]$ to the partial transportability task defined over discrete SCMs is a tight bound over a target query $\mathbb{E}_{P*}[Y \mid \mathbf{x}]$. The credible interval $[\hat{l}_0(\mathbf{x}), \hat{u}_0(\mathbf{x})]$ coincides with $[l(\mathbf{x}), u(\mathbf{x})]$ as $n_i \to \infty$ in all observable domains $\pi^i$, $i = 1, \ldots, k$.*

## 5 EXPERIMENTS

### 5.1 SMOKING AND LUNG CANCER

Our first experiment is inspired by the debate around the relationship between smoking and lung cancer in the 1950's (US Department of Health and Human Services, 2014). We use a scientifically-grounded variation of the front-door graph that includes an individual's smoking status $S$, presence of tar in the lungs $T$, wealth $W$, and lung cancer status $C$, using the fact that smoking and lung cancer may be confounded by an individual's unobserved genetic profile. In this example, the objective is to make inference on cancer probability distributions in the French population $\pi^{\text{FR}}$ from corresponding data in $\pi^{\text{UK}}$ where the prevalence of smoking is known to be lower. The selection diagram is given in Fig. 3a and details on the SCMs used to generate data are given in Appendix B.

**Probability of cancer among smokers $P^{\text{FR}}(C = 1 \mid S = 1)$.** The $C$-factor decomposition and parameterization is given by the following derivations,

$$P^{\text{FR}}(c \mid s) = \frac{P^{\text{FR}}(c, s)}{\sum_c P^{\text{FR}}(c, s)} = \frac{\sum_{t,w} P^{\text{FR}}(c, s, t, w)}{\sum_{c,t,w} P^{\text{FR}}(c, s, t, w)} = \frac{\sum_{t,w} Q^{\text{FR}}[s, c] Q^{\text{FR}}[w] Q^{\text{FR}}[t]}{\sum_{c,t,w} Q^{\text{FR}}[s, c] Q^{\text{FR}}[w] Q^{\text{FR}}[t]}, \tag{16}$$

where $Q^{\text{FR}}[t] = Q^{\text{UK}}[t] = P^{\text{UK}}(t \mid s, w)$ and $Q^{\text{FR}}[w] = Q^{\text{UK}}[w] = P^{\text{UK}}(w)$, and,

$$Q^{\text{FR}}[s, c] = \sum_{u_{sc}, u_s} \mathbb{1}\{\xi_{C^{\text{UK}}}^{(w, t, u_{sc})} = c\} \mathbb{1}\{\xi_{S^{\text{FR}}}^{(w, u_{sc}, u_s)} = s\} \theta_{u_{sc}}^{\text{UK}} \theta_{u_s}^{\text{FR}}.$$

In Fig. 4, we report estimated 100% credible intervals $\hat{l}_0 < P^{\text{FR}}(C = 1 \mid S = 1) < \hat{u}_0$ as a function of the number of samples without prior information (purple) and with the prior information that $P^{\text{FR}}(s \mid w)$ lies in an interval of width 0.1 around its true value (pink). The black and gray dotted lines are the actual values

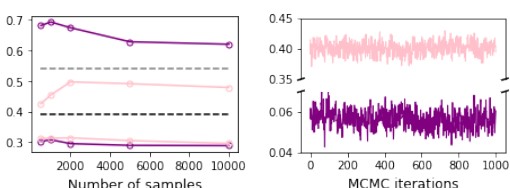

Figure 4: Bounding the probability of cancer.

$P^{\text{FR}}(C = 1 \mid S = 1)$ and $P^{\text{UK}}(C = 1 \mid S = 1)$ respectively. Notice that a relatively small number of samples is required to converge to stable bounds, and that the prior information narrows the credible interval reflecting this additional constraint. We also show for illustration that our Gibbs sampler recovers the true values $P^{\text{FR}}(C = 1 \mid S = 1)$ (pink) and $P^{\text{FR}}(C = 1 \mid S = 0)$ (purple) when trained on data from $\pi^{\text{FR}}$, i.e. when probabilities are identified.

**Prediction performance across domains.** Consider the task of designing cancer prediction rules for optimal performance in the french population $\pi^{\text{FR}}$. We introduce an additional training domain to be able to define invariant predictors: data from the Swedish population $\pi^{\text{SW}}$ whose structural differences with $\pi^{\text{UK}}$ and with $\pi^{\text{FR}}$ are given in Fig. 3. Across $\pi^{\text{UK}}$ and $\pi^{\text{SW}}$, the optimal invariant predictor (Def. 3) is given by $\mathbb{E}_{P^{\text{UK}}}[C \mid t, w, s] = \mathbb{E}_{P^{\text{SW}}}[C \mid t, w, s]$ which, however, is not equal to $\mathbb{E}_{P^{\text{FR}}}[C \mid t, w, s]$ as no set blocks the open path between the selection node $S_S$ and the cancer variable $C$ in $\mathcal{G}^{\text{FR,UK}}$. We consider also the common strategy of using causal parents for prediction, i.e. using the prediction rule $\mathbb{E}_{P^{\text{UK}}}[C \mid t, w]$ (which, similarly, is not equal to $\mathbb{E}_{P^{\text{FR}}}[C \mid t, w]$). For comparison, we consider the median value $\text{med}(\hat{l}_0, \hat{u}_0)$ for the optimal prediction rule $\mathbb{E}_{P^{\text{FR}}}[C \mid t, w, s]$ computed using data from $\pi^{\text{UK}}$ and $\pi^{\text{SW}}$. We observe in Fig. 3d that indeed the prediction rule $\mathbb{E}_{P^{\text{UK}}}[C \mid t, w, s]$ underperforms in $\pi^{\text{FR}}$: for reference $\mathbb{E}_{P^{\text{FR}}}[C \mid t, w, s]$ has a mean error of .1220, cautioning against naively transporting invariant prediction rules across domains. Similarly, using causal parents for prediction underperforms. In contrast, the median of the derived bound proves to be a slightly better predictor in this case and has a guarantee of optimal performance in the "worst-case" domain compatible with the selection diagrams (Prop. 4).

## 5.2 PREDICTION OF NEURODEGENERATIVE DISEASES ACROSS HOSPITALS

Our second experiment reconsiders the introductory example that described the design of prediction rules for the development of neurodegenerative diseases in a target hospital $\pi^*$ in which no data has been recorded. Instead, we have access to data from two related studies conducted in hospitals $\pi^a$ and $\pi^b$ that, however, are known to differ with respect to the target domain notably in the distribution of blood pressure $W$, a known cause of neurodegenerative diseases. The causal protocol is given in Fig. 1 and is described in more depth in Sec. 2. Details on the SCMs used to generate data are given in Appendix B.

Given this information, we consider the task of designing a prediction rule for optimal mean squared error in the target hospital $\pi^*$. Here, invariant predictors are well defined and given by the function $f(w, x_1, x_2) = \mathbb{E}_{P^a}[Y \mid w, x_1, x_2] = \mathbb{E}_{P^b}[Y \mid w, x_1, x_2]$ although note that, in this example, this conditional expectation is not invariant in the target domain due to the difference in the causal mechanisms associated with blood pressure $W$, see Fig. 1d. Similarly, we can define causal predictors as $\mathbb{E}_{P^a}[Y \mid w, x_2]$ and $\mathbb{E}_{P^b}[Y \mid w, x_2]$ which in this case are not equal across hospitals $\pi^a$ and $\pi^b$ due to the open path between $S_{X_1}$ and $Y$ once we condition on $W$. The partial transportability task instead argues for approximating $\mathbb{E}_{P^*}[Y \mid w, x_1, x_2]$ which, using the $C$-factor decomposition, is parameterized by $P^*(y, w, x_1, x_2) = Q^*[X_1, X_2]Q^*[W, Y]$ where $Q^*[X_1, X_2] = P^a(x_1, x_2)$ by Lem. 1 and

$$Q^*[W, Y] = \sum_{u_{wy}, u_w} \mathbb{1}\{\xi_{Y^a}^{(w, x_2, u_{wy})} = y\}\mathbb{1}\{\xi_{W*}^{(x_1, u_{wy}, u_w)} = w\}\theta_{u_{wy}}^a \theta_{u_w}^*.$$

The median value of the resulting interval that encodes the uncertainty in the computation of $\mathbb{E}_{P^*}[Y \mid w, x_1, x_2]$ as well as all baseline predictors are given in Fig. 5. We add the actual optimal (not computable) prediction rule $\mathbb{E}_{P^*}[Y \mid w, x_1, x_2]$ for reference. Fig. 5 shows that the median outperforms and that baselines, although common strategies for prediction, can result in significantly worse out-of-distribution performance in examples where unobserved confounding as well as structural differences between domains play a role.

|  | $\pi^*$ |
|---|---|
| $\mathbb{E}_{P^a}[y\|w, x_1, x_2]$ | .3640 (.003) |
| $\mathbb{E}_{P^a}[y\|w, x_2]$ | .4244 (.002) |
| $\mathbb{E}_{P^b}[y\|w, x_2]$ | .4013 (.002) |
| $\text{med}(\hat{l}_0, \hat{u}_0)$ | .2961 (.008) |
| $\mathbb{E}_{P^*}[y\|w, x_1, x_2]$ | .2434 (.002) |

Figure 5: Performance comparisons.

## 6 CONCLUSIONS

This paper investigated the problem of domain generalization from the perspective of transportability theory. We introduced the task of partial transportability that seeks to bound the value of an arbitrary

conditional expectation $\mathbb{E}_{P*}[Y \mid \mathbf{x}]$ in an unseen domain $\pi^*$ using selection diagrams and data from source domains. Using this formalism, we showed that invariant predictors and more general solutions to robust optimization problems derived in the literature are special cases of solutions to this task. Moreover, in systems of discrete observables, we showed that we can design provably consistent algorithms for inferring bounds that are sound and tight, and illustrated its performance on synthetic data.

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

# A PROOFS

**Proposition 6** (Prop. 1 restated). *Given selection diagrams $\{\mathcal{G}^{i,j} : i, j = 1, \ldots, k\}$, the optimal invariant predictor is defined as the minimizer of $\mathbb{E}_{P^i}[(Y - f(\mathbf{Z}))^2]$ across all $i = 1, \ldots, k$, and belongs to the set of invariant predictors for which $\mathbf{Z}$ is a domain-independent Markov blanket for $Y \in \mathbf{V}$.*

*Proof.* Assume not such that there exists an optimal invariant predictor $\mathbb{E}_{P^i}[Y \mid \mathbf{Z}], i = 1, \ldots, k$, distinct from any invariant predictor defined conditional on a domain-independent Markov blanket, with $\mathbf{Z}$ not a domain-independent Markov blanket for $Y \in \mathbf{V}$. Then, by definition of a domain-independent Markov blanket, either $(Y \perp\!\!\!\perp \mathbf{S} \mid \mathbf{Z})_{\mathcal{G}^{i,j}}$ for $i, j = 1, \ldots, k$, in which case $\mathbb{E}_{P^i}[Y \mid \mathbf{Z}]$ is not invariant across source domains, or there exists a $W \in \mathbf{Z}$ such that $(W \perp\!\!\!\perp Y \mid \mathbf{Z}\backslash W)_{\mathcal{G}^{i,j}}$ for $i, j = 1, \ldots, k$, in which case $\mathbb{E}_{P^i}[Y \mid \mathbf{Z}] = \mathbb{E}_{P^i}[Y \mid \mathbf{Z}\backslash W]$. Now if $\mathbf{Z}\backslash W$ is not a domain-independent Markov blanket we can continue removing independent variables from $\mathbf{Z}\backslash W$ to reach a domain-independent Markov blanket concluding that $\mathbb{E}_{P^i}[Y \mid \mathbf{Z}]$ is not distinct from an invariant predictor defined conditional on a domain-independent Markov blanket. $\square$

**Proposition 7** (Prop. 2 restated). *Given a set of selection diagrams $\{\mathcal{G}^{i,j} : i, j = 1, \ldots, k\}$, let $\Delta = \bigcup_{i,j} \Delta^{i,j}$ be the set of variables in $\mathbf{V}$ whose causal mechanisms differ between any two source domains, and let $\mathbf{S} = \{S_V : V \in \Delta\}$. The optimal invariant predictor solves the robust optimization problem in Eq. (2) if selection nodes in all selection diagrams $\{\mathcal{G}^{i,*} : i = 1, \ldots, k\}$ are given by $\mathbf{S}$ with edges $\{S_V \to V\}_{S_V \in \mathbf{S}}$.*

*Proof.* Given a set of selection diagrams $\{\mathcal{G}^{i,j} : i, j = 1, \ldots, k\}$, let $\Delta = \bigcup_{i,j} \Delta^{i,j}$ be the set of variables in $\mathbf{V}$ whose causal mechanisms differ between any two source domains, and let $\mathbf{S} = \{S_V : V \in \Delta\}$.

Assume that selection nodes in all selection diagrams $\{\mathcal{G}^{i,*} : i = 1, \ldots, k\}$ are given by $\mathbf{S}$ (and with edges $\{S_V \to V\}_{S_V \in \mathbf{S}}$). In that case, the optimal invariant predictor, written $\mathbb{E}_{P^1}[Y \mid \mathbf{Z}] = \mathbb{E}_{P^M}[Y \mid \mathbf{Z}]$ for any $M \in \mathcal{M}(\mathcal{G})$. Any additional variable $W$ in the conditioning set is either irrelevant for prediction, i.e. $\mathbb{E}_{P^M}[Y \mid \mathbf{Z}] = \mathbb{E}_{P^M}[Y \mid \mathbf{Z}, W]$, or breaks the independence between $Y$ and selection nodes, which implies that $\mathbb{E}_{P^M}[Y \mid \mathbf{Z}, W]$ varies as a function of $M$. Since the the functional form of $M$ (besides the arguments of functions) are not constrained by selection diagrams, for any fixed prediction function $\mathbb{E}_{P^M}[Y \mid \mathbf{Z}, W]$, we can always find a domain $M' \in \mathcal{M}(\mathcal{G})$ that makes the error $\mathbb{E}_{P^{M'}}[(Y - \mathbb{E}_{P^M}[Y \mid \mathbf{Z}, W])^2]$ arbitrarily large, and thus higher than $\mathbb{E}_{P^{M'}}[(Y - \mathbb{E}_{P^M}[Y \mid \mathbf{Z}])^2]$ which is fixed for any $M'$. $\square$

**Proposition 8** (Prop. 3 restated). *For a given set of selection diagrams, let $[l(\mathbf{x}), u(\mathbf{x})]$ denote the solution of the partial transportability task for the query $\mathbb{E}_{P^M}[Y \mid \mathbf{x}], M \in \mathcal{M}(\mathcal{G}^*)$ and $\mathbb{E}_{P^1}[Y \mid \mathbf{z}], \mathbf{Z} \subseteq \mathbf{X}$ be the invariant predictor. Then, $\mathbb{E}_{P^1}[Y \mid \mathbf{z}] \in [l(\mathbf{x}), u(\mathbf{x})]$. Moreover, by definition $\mathbb{E}_{P^M}[Y \mid \mathbf{x}] \in [l(\mathbf{x}), u(\mathbf{x})]$ for a particular "worst-case" member $M \in \mathcal{M}(\mathcal{G}^*)$.*

*Proof.* The set $\mathcal{M}(\mathcal{G}^*)$ represents all SCMs compatible with a target causal graph $\mathcal{G}^*$ that is only constrained by selection diagrams $\{\mathcal{G}^{i,*} : i = 1, \ldots, k\}$. A selection node indicates a potential change between two domains and therefore, in principle all source domain SCMs $\{M^i : i = 1, \ldots, k\}$ are possible candidates for the target domain and thus $M^i \in \mathcal{M}(\mathcal{G}^*)$ $i = 1, \ldots, k$. Then, by the definition of the partial transportability task, $\mathbb{E}_{P^1}[Y \mid \mathbf{z}] \in [l(\mathbf{x}), u(\mathbf{x})]$ where $\mathbf{z}$ is the value of $\mathbf{Z}$ in $\mathbf{X}$. $\square$

**Proposition 9** (Prop. 4 restated). *For a given set of selection diagrams and data, let $[l(\mathbf{x}), u(\mathbf{x})]$ denote the solution of the partial transportability task for the query $\mathbb{E}_{P^M}[Y \mid \mathbf{x}], M \in \mathcal{M}(\mathcal{G}^*)$. Then,*

$$\max_{M \in \mathcal{M}(\mathcal{G}^*)} \mathbb{E}_{P^M}\left[(Y - \underset{M \in \mathcal{M}(\mathcal{G}^*)}{med} \mathbb{E}_{P^M}[Y \mid \mathbf{X}])^2\right]$$

$$\leq \max_{M \in \mathcal{M}(\mathcal{G}^*)}\left(\mathbb{E}_{P^M}[(Y - \mathbb{E}_{P^M}[Y \mid \mathbf{X}])^2] + \frac{1}{4}\mathbb{E}_{P^M}[(u(\mathbf{X}) - l(\mathbf{X}))^2]\right).$$

*Under the condition that the irreducible error $\mathbb{E}_{P^M}[(Y - \mathbb{E}_{P^M}[Y \mid \mathbf{X}])^2]$ is constant across $M \in \mathcal{M}(\mathcal{G}^*)$, $med_{M \in \mathcal{M}(\mathcal{G}^*)}\mathbb{E}_{P^M}[Y \mid \mathbf{X}]$ provably solves the robust optimization problem Eq. (2).*

*Proof.* For a given set of selection diagrams and data, let $[l(\mathbf{x}), u(\mathbf{x})]$ denote the solution of the partial transportability task for the query $\mathbb{E}_{PM}[Y \mid \mathbf{x}]$, $M \in \mathcal{M}(\mathcal{G}^*)$. Then,

$$\max_{M \in \mathcal{M}(\mathcal{G}^*)} \mathbb{E}_{PM}[(Y - \underset{M \in \mathcal{M}(\mathcal{G}^*)}{\mathrm{med}} \mathbb{E}_{PM}[Y \mid \mathbf{X}])^2]$$

$$= \max_{M \in \mathcal{M}(\mathcal{G}^*)} \left( \mathbb{E}_{PM}\left[ \left( Y - \mathbb{E}_{PM}[Y \mid \mathbf{X}] + \mathbb{E}_{PM}[Y \mid \mathbf{X}] - \underset{M \in \mathcal{M}(\mathcal{G}^*)}{\mathrm{med}} \mathbb{E}_{PM}[Y \mid \mathbf{X}] \right)^2 \right] \right)$$

$$= \max_{M \in \mathcal{M}(\mathcal{G}^*)} \left( \mathbb{E}_{PM}[(Y - \mathbb{E}_{PM}[Y \mid \mathbf{X}])^2] + \mathbb{E}_{PM}[(\mathbb{E}_{PM}[Y \mid \mathbf{X}] - \underset{M \in \mathcal{M}(\mathcal{G}^*)}{\mathrm{med}} \mathbb{E}_{PM}[Y \mid \mathbf{X}])^2] \right)$$

$$\leqslant \max_{M \in \mathcal{M}(\mathcal{G}^*)} \left( \mathbb{E}_{PM}[(Y - \mathbb{E}_{PM}[Y \mid \mathbf{X}])^2] + \frac{1}{4} \mathbb{E}_{PM}[(u(\mathbf{X}) - l(\mathbf{X}))^2] \right).$$

The second equality holds because the cross term in the expansion of the square equal 0 as $\mathbb{E}_{PM}[(Y - \mathbb{E}_{PM}[Y \mid \mathbf{X}])] = 0$. The inequality holds because the largest distance between $\mathbb{E}_{PM}[Y \mid \mathbf{X}]$ and the median of values $\mathbb{E}_{PM}[Y \mid \mathbf{X}]$ can reach as a function of $M \in \mathcal{M}(\mathcal{G}^*)$ is half the distance between maximum and minimum values of $\mathbb{E}_{PM}[Y \mid \mathbf{X}]$ across $M \in \mathcal{M}(\mathcal{G}^*)$, that is $(u(\mathbf{X}) - l(\mathbf{X}))/2$.

If $\mathbb{E}_{PM}[(Y - \mathbb{E}_{PM}[Y \mid \mathbf{X}])^2]$ is equal to a constant value independent of $M$, it can be taken out of the maximization and we are left with the optimization problem,

$$\arg\min_{f} \max_{M \in \mathcal{M}(\mathcal{G}^*)} \mathbb{E}_{PM}[(\mathbb{E}_{PM}[Y \mid \mathbf{X}] - f(\mathbf{X}))^2] \tag{17}$$

For any $x$ and any $f$, we can always choose $M$ such that $|\mathbb{E}_{PM}[Y \mid \mathbf{x}] - f(\mathbf{x})| \geqslant |\mathbb{E}_{PM}[Y \mid \mathbf{x}] - \underset{M \in \mathcal{M}(\mathcal{G}^*)}{\mathrm{med}} \mathbb{E}_{PM}[Y \mid \mathbf{x}]|$. For example, by choosing $M$ such that $\mathbb{E}_{PM}[Y \mid \mathbf{x}] = \max_{M \in \mathcal{M}(\mathcal{G}^*)} \mathbb{E}_{PM}[Y \mid \mathbf{x}]$ or $\mathbb{E}_{PM}[Y \mid \mathbf{x}] = \min_{M \in \mathcal{M}(\mathcal{G}^*)} \mathbb{E}_{PM}[Y \mid \mathbf{x}]$ depending on what distance is larger. Therefore, $f(\mathbf{x}) := \underset{M \in \mathcal{M}(\mathcal{G}^*)}{\mathrm{med}} \mathbb{E}_{PM}[Y \mid \mathbf{x}]$ minimizes the robust optimization problem. $\qquad \square$

**Theorem 2** (Prop. 5 restated). *Let $M$ be an arbitrary SCM with graph $\mathcal{G}$ and let $\mathbf{C}$ be any c-component. Then, there exists a corresponding SCM $N$ with finite exogenous domain compatible with $\mathcal{G}$ such that $Q_M[\mathbf{C}] = Q_N[\mathbf{C}]$, where for every exogenous variable $U \in \mathbf{U_C}$, its cardinality $d_U = |\Omega_{Pa(\mathbf{C})}|$.*

*Proof.* The proof follows from Rosset et al. (2017) and Zhang et al. (2021). We include it below for completeness.

We first introduce some necessary notations and concepts. The probability distribution for every exogenous variables $U \subset \mathbf{U}$ is characterized with a probability space. It is frequently designated $\langle \Omega_U, \mathcal{F}_U, P_U \rangle$ where $\Omega_U$ is a sample space containing all possible outcomes; $\mathcal{F}_U$ is a $\sigma$-algebra containing subsets of $\Omega_U$; $P_U$ is a probability measure on $\mathcal{F}_U$ normalized such that $P_U(\Omega_U) = 1$. Elements of $\mathcal{F}_U$ are called events, which are closed under operations of set complement and unions of countably many sets. By means of $P_U$ a real number $P_U(\mathcal{A}) \in [0, 1]$ is assigned to every event $\mathcal{A} \in \mathcal{F}_U$; it is called the probability of event $\mathcal{A}$. For an arbitrary set of exogenous variables $\mathbf{U}$, its realization $\mathbf{U} = \mathbf{u}$ is an element in the Cartesian product $\times_{U \in \mathbf{U}} \Omega_U$. We may be interested in inferring whether a sequence of events $\mathcal{A}$ for every $U \in \mathbf{U}$ occurs. Such an event is represented by a subset $\times_{U \in \mathbf{U}} \mathcal{A}_U \subseteq \times_{U \in \mathbf{U}} \Omega_U$ which in turn generate a product of $\sigma$-algebras $\bigotimes_{U \in \mathbf{U}} \mathcal{F}_U$. Define the product measure $\bigotimes_{U \in \mathbf{U}} P_U$ to satisfy the following mutual independence condition given by the definition of the SCM,

$$P\left( \underset{U \in \mathbf{U}}{\times} \mathcal{A}_U \right) = \prod_{U \in \mathbf{U}} P_U(\mathcal{A}_U). \tag{18}$$

Such $P$ is a probability measure. Moreover,

$$\left\langle \underset{U \in \mathbf{U})}{\times} \Omega_U, \bigotimes_{U \in \mathbf{U}} \mathcal{F}_U, \bigotimes_{U \in \mathbf{U}} P_U \right\rangle, \tag{19}$$

defines a product of probability spaces $\langle \Omega_U, \mathcal{F}_U, P_U \rangle$ that describes measurable events over all exogenous variables $\mathbf{U}$ partitioned into $c$-components.

Let $\mathbb{C}$ be the collection of all $c$-components in $\mathcal{G}$. $c$-components in $\mathbb{C}$ form a partition $\{\bigcup_{V \in \mathbf{C}} U_V \mid \mathbf{C} \in \mathbb{C}\}$ over exogenous variables $U$. Therefore, for every $U \in \mathbf{U}$, there must exist a unique $c$-component denoted by $\mathbf{C}_U$ containing $U$. For any $c$-component $\mathbf{C} \in \mathbb{C}$, let $U_{\mathbf{C}} = \bigcup_{V \in \mathbf{C}} U_V$ the set of exogenous variables affecting (at least one of) endogenous variables in $\mathbf{C}$. By the definition of $c$-components, the exogenous variables do not overlap between $c$-components and it holds that,

$$P\left(\bigcap_{U \in \mathbf{U}} \mathcal{A}_U\right) = \prod_{C \in \mathcal{C}(\mathcal{G})} P_U\left(\bigcap_{U \in C} \mathcal{A}_U\right). \tag{20}$$

For any SCM $M$ compatible with the causal graph $\mathcal{G}$ the joint distribution may be factorized into $c$-components,

$$P(\mathbf{v}) = \prod_{\mathbf{C} \in \mathbb{C}} Q[\mathbf{C}](\mathbf{c}, \boldsymbol{pa}_{\mathbf{C}}). \tag{21}$$

where $Q[\mathbf{C}]$ is a $C$-factor and is a function of $(\mathbf{c}, \boldsymbol{pa}_{\mathbf{C}})$.

To parameterize this joint distribution it is thus sufficient to look at each $C$-factor separately. Let $\mathbf{C}$ be a generic $c$-component in $\mathcal{G}$. Denote by $m = |U_{\mathbf{C}}|$ the number of exogenous variables related to $\mathbf{C}$. For convenience, we consistently write $\langle \Omega_i, \mathcal{F}_i, P_i \rangle$ as the probability space of $i$-th exogenous variable in $C$. The product of these probability spaces is thus written,

$$\left\langle \underset{i=1}{\overset{m}{\times}} \Omega_i, \bigotimes_{i=1}^{m} \mathcal{F}_i, \bigotimes_{i=1}^{m} P_i \right\rangle. \tag{22}$$

Each $C$-factor may thus be written,

$$Q[\mathbf{C}] = \int_{\times_{i=1}^{m} \Omega_i} \prod_{V \in \mathbf{C}} \mathbb{1}\{f_V(\boldsymbol{pa}_V, \boldsymbol{u}_V) = v\} d\bigotimes_{i=1}^{m} P_i. \tag{23}$$

Our goal is to show that all probabilities $Q[\mathbf{C}]$, induced by exogenous variables described by arbitrary probability spaces could be produced by a "simpler" generative process with discrete exogenous domains. $Q[\mathbf{C}]$ defines a mapping between the space of possible realizations of the variables $Pa(\mathbf{C})$ to the $[0, 1]$ interval. Since $Pa(\mathbf{C})$ are discrete variables with finite domains, the cardinality of the class of probability assignments that must be defined is also finite. It is given at most by the number of possible combinations of realizations of $Pa(\mathbf{C})$ which is given by $\prod_{V \in Pa(\mathbf{C})} |\Omega_V|$.

Let $\bar{P}$ be a vector representing probabilities $Q[\mathbf{C}](\mathbf{c}, \boldsymbol{pa}_{\mathbf{C}})$. Counting all possible combinations of outcomes for all possible conditioning sets, $\bar{P}$ is therefore a vector of at most size $d = \prod_{V \in Pa(\mathbf{C})} |\Omega_V|$. And since $Q[\mathbf{C}](\mathbf{c}, \boldsymbol{pa}_{\mathbf{C}})$ is a probability mass function, it only takes a vector with $d - 1$ dimensions to uniquely determine it. $\bar{P}$ may thus be interpreted as a point in the $(d-1)$-dimensional real space. Similarly, $(P, 1)$ is vector in $d$-dimensional space where the $d$-th element is equal to 1.

Now consider sampling a value $U_1 = u_1$ from the underlying SCM and let $Q_{u_1}$ be the probability model with $U_1 = u_1$.

$$Q_{u_1}[\mathbf{C}](\mathbf{c} \mid \boldsymbol{pa}_{\mathbf{C}}) = \left[\int_{\times_{i=2}^{m} \Omega_i} \prod_{V \in \mathbf{C}} \mathbb{1}\{f_V(\boldsymbol{pa}_V, \boldsymbol{u}_V) = v\} d\bigotimes_{i=2}^{m} P_i\right]_{U_1 = u_1}. \tag{24}$$

and $\bar{P}_{u_1}$ is a $(d-1)$-dimensional probability vector representing the probabilities of each one of the combinations $Pa(\mathbf{C})$ given that $U_1 = u_1$. We will show that $P_1$ may equally well be represented by a discrete distribution. For this, let $\mathcal{U} = \{\bar{P}_{u_1} : u_1 \in \Omega_1\} \subset \mathbb{R}^d$ be the set of probability points that can be constructed as $u_1$ varies in $\Omega_1$. The average $\int_{\Omega_1} \bar{P}_{u_1} dP_1$ is a convex mixture of points in $\mathcal{U}$ by (Rubin & Wesler, 1958) that equals $\bar{Q}$ since,

$$\bar{P} = \int_{\Omega_1} \left[\int_{\times_{i=2}^{m} \Omega_i} \prod_{V \in \mathbf{C}} \mathbb{1}\{f_V(\boldsymbol{pa}_V, \boldsymbol{u}_V) = v\} d\bigotimes_{i=2}^{m} P_i\right]_{U_1 = u_1} dP_1. \tag{25}$$

By construction, $\bar{P}$ itself is a convex mixture of at most $d + 1$ points in $\mathcal{U}$. That is, by using Carathéodory's theorem (Carathéodory, 1911),

$$\bar{P} = \sum_{k=1}^{d+1} w_k \bar{P}_{u_{1,k}}. \tag{26}$$

Replacing the definition of $\bar{P}_{u_{1,k}}$ we obtain,

$$\bar{P} = \sum_{k=1}^{d+1} w_k \left[ \int_{\times_{i=2}^m \Omega_i} \prod_{V \in \mathbf{C}} \mathbb{1}\{f_V(\boldsymbol{pa}_V, \boldsymbol{u}_V) = v\} d \bigotimes_{i=2}^m P_i \right]_{U_1 = u_{1,k}} \tag{27}$$

This means that we can replace the continuous measure $P_1$ with a discrete probability set with outcomes $\{u_{1,1}, \ldots, u_{1,d}\}$ and corresponding probabilities $\{w_1, \ldots, w_d\}$ with cardinality $d$ and obtain a probability model that is equivalent to the original $\bar{P}$. This procedure can be repeated for all $m$ exogenous variables in the $c$-component $\mathbf{C}$. We are thus left with a model,

$$Q[\mathbf{C}](\mathbf{c}, \boldsymbol{pa}_{\mathbf{C}}) = \int_{\times_{i=1}^m \Omega_i} \prod_{V \in \mathbf{C}} \mathbb{1}\{f_V(\boldsymbol{pa}_V, \boldsymbol{u}_V) = v\} d \bigotimes_{i=1}^m P_i, \tag{28}$$

equivalent to its discrete counterpart,

$$Q[\mathbf{C}](\mathbf{c}, \boldsymbol{pa}_{\mathbf{C}}) = \sum_{u \in \mathbf{U}_{\mathbf{C}}} \sum_{u=1,\ldots,d} \prod_{V \in \mathbf{C}} \mathbb{1}\{f_V(\boldsymbol{pa}_V, \boldsymbol{u}_V) = v\} \prod_{u \in \boldsymbol{u}_{\mathbf{c}}} P(u), \tag{29}$$

where $d = \prod_{V \in Pa(\mathbf{C})} |\Omega_V|$.

$\square$

**Theorem 3** (Thm. 1 restated). *The solution $[l, u]$ to the partial transportability task defined over discrete SCMs is a tight bound over a target query $\mathbb{E}_{P_{\pi*}}[y \mid \mathbf{x}]$. The credible interval $[\hat{l}_0, \hat{u}_0]$ coincides with $[l, u]$ as $n_i \to \infty$ in all observable domains $\pi^i$, $i = 1, \ldots, k$.*

*Proof.* The proof strategy follows (Zhang et al., 2021) and shows convergence of the posterior by way of convergence of the likelihood of the data given one SCM $M \in \mathcal{M}(\mathcal{G})$. We look at 'convergence' in a frequentist way, for increasing sample size the posterior will, with increasing probability, be low for any parameter configuration, i.e. for any SCM $M \notin \mathcal{M}(\mathcal{G})$.

By the definition of the optimal bounds $[l, u]$ given by the solution to the partial transportability task,

$$P(\bar{\mathbf{v}} \mid l < \mathbb{E}_{P_{\pi*}}[y \mid \mathbf{x}] < u) \to_p 1. \tag{30}$$

Therefore if the prior on parameters $(\boldsymbol{\xi}, \boldsymbol{\theta})$ defining SCMs is non-zero for any $M \in \mathcal{M}(\mathcal{G})$, also the posterior converges,

$$P(l < \mathbb{E}_{P_{\pi*}}[y \mid \mathbf{x}] < u \mid \bar{\mathbf{v}}) \to_p 1, \tag{31}$$

which is the definition credible intervals $[l_0, u_0]$ as the $0^{th}$ and $100^{th}$ quantiles of the posterior distribution which coincide with $[l, u]$ asymptotically.

$\square$

## B    EXPERIMENTAL DETAILS

All experiments use 1000 burn-in MCMC samples that are discarded and 5000 MCMC samples considered as independent samples from the posterior distribution and used for the approximation of target queries.

## B.1 SCMs for the smoking and lung cancer example

Using the functional dependencies specified by the selection diagram in Fig. 4, we define the SCMs for domains $\pi^{\text{UK}}$, $\pi^{\text{FR}}$, and $\pi^{\text{SW}}$ as follows.

For $\pi^{\text{UK}}$ we generate samples from $P(u_w), P(u_s), P(u_t)$ and $P(u_{sc})$ given by independent Gaussian distributions with mean 0 and variance 1. Each generated $(u_w, u_s, u_t, u_{sc})$ leads to a sample $(w, s, t, c)$ as follows: $w \leftarrow \mathbb{1}\{u_w > 0\}, s \leftarrow \mathbb{1}\{w + u_{sc} + u_s - 2 > 0\}, t \leftarrow \mathbb{1}\{s - 0.5u_t - 1 > 0\}, c \leftarrow \mathbb{1}\{t - 0.5w + u_{sc} - 1 > 0\}$.

For $\pi^{\text{FR}}$ we generate samples from $P(u_w), P(u_s), P(u_t)$ and $P(u_{sc})$ given by independent Gaussian distributions with mean 0 and variance 1. Each generated $(u_w, u_s, u_t, u_{sc})$ leads to a sample $(w, s, t, c)$ as follows: $w \leftarrow \mathbb{1}\{u_w > 0\}, s \leftarrow \mathbb{1}\{w + u_{sc} + 1.5u_s - 1 > 0\}, t \leftarrow \mathbb{1}\{s - 0.5u_t - 1 > 0\}, c \leftarrow \mathbb{1}\{t - 0.5w + u_{sc} - 1 > 0\}$. Notice that the causal mechanism for $S$ has changed while everything else is unchanged.

For $\pi^{\text{SW}}$ we generate samples from $P(u_w), P(u_s), P(u_t)$ and $P(u_{sc})$ given by independent Gaussian distributions with mean 0 and variance 1. Each generated $(u_w, u_s, u_t, u_{sc})$ leads to a sample $(w, s, t, c)$ as follows: $w \leftarrow \mathbb{1}\{u_w > 0.5\}, s \leftarrow \mathbb{1}\{w + u_{sc} + u_s - 2 > 0\}, t \leftarrow \mathbb{1}\{s - 0.5u_t - 1 > 0\}, c \leftarrow \mathbb{1}\{t - 0.5w + u_{sc} - 1 > 0\}$.

## B.2 SCMs for the neurodegenerative disease prediction example

Using the functional dependencies specified by the selection diagrams in Fig. 1, we define the SCMs for domains $\pi^*$, $\pi^a$, and $\pi^b$ as follows.

For the target domain $\pi^*$ we generate samples from $P(u_{wy}), P(u_{x_2}, P(u_w)$ and $P(u_{x_1,x_2})$ given by independent Gaussian distributions with mean 0 and variance 1. Each generated $(u_{wy}, u_{x_1,x_2}, u_{x_2}, u_w)$ leads to a sample $(x_1, x_2, w, y)$ as follows: $x_1 \leftarrow \mathbb{1}\{u_{x_1} > 0\}, x_2 \leftarrow \mathbb{1}\{u_{x_1,x_2} + u_{x_2} > 0\}, w \leftarrow \mathbb{1}\{x_1 + u_{wy} + 1.5u_w - 1 > 0\}, y \leftarrow \mathbb{1}\{w - u_{wy} + 0.1x_1 - 1 > 0\}$.

For source domain $\pi^a$, the distribution of exogenous as well as structural assignment agree with $\pi^*$ except in the assignment of $W$ which is given by $w \leftarrow \mathbb{1}\{x_1 + u_{wy} - u_w + 1 > 0\}$.

For source domain $\pi^b$, the distribution of exogenous as well as structural assignment agree with $\pi^*$ except in the assignment of $W$ and $X_1$. The selection diagram specifies that the assignment of $W$ agrees with $\pi^a$ and is thus given by $w \leftarrow \mathbb{1}\{x_1 + u_{wy} - u_w + 1 > 0\}$ while the assignment of $X_1$ changes and is given by $x_1 \leftarrow \mathbb{1}\{u_{x_1} - 0.5 > 0\}$. All other components of the SCM are the same.

