# OpenReview forum: "Partial transportability for domain generalization"
_ICLR.cc/2023/Conference — Submitted to ICLR 2023_

### Official Review · Reviewer_D5NR · 2022-10-24

**Confidence:** 4
**Correctness:** 2
**Technical Novelty And Significance:** 2
**Empirical Novelty And Significance:** 2
**Recommendation:** 3

**Clarity, Quality, Novelty And Reproducibility:**

Overall, the paper is not hard to follow. The results are clearly presented in terms of the writing itself. The proofs for theoretical results and detailed settings in the empirical results are provided in the appendix.

**Strength And Weaknesses:**

## Strength

The strength of the paper originates from the effort of trying to analyze a specific type of domain generalization problem via transportability modeling. I believe the utilization of the appropriate causal modeling (if feasible) is of crucial importance when pursuing principled analysis towards domain generalization, and the paper makes an effort along this direction.

## Weakness

The weakness of the paper results from the lack of clarity of the definition and configuration, as well as the worry about the overall significance of the results.

### 1. w.r.t. Definition 2: Selection diagram

Selection diagram is the core definition with respect to which major theoretical results in the paper are presented. While I agree with authors that we need some structural assumptions on domain generalization so that we can perform analyses, I am hesitant to agree with the way assumptions are introduced with "selection diagram". In my opinion, the definition is confusing. If the $S$ variable is an additional cause for the observed variable (e.g., Figure 1b - 1d), this is essentially changing the underlying causal semantics among the observed variables, which is not the setting described in the paper. In the medical study example in Section 2, what is the additional cause for $W$ (Figure 1b) in this context? Shouldn't it be $W \rightarrow S_W$ since it is a (general) kind of selection? Figure 1c - 1d share the same problem.

### 2. the modeling of domain generalization

As a follow-up to 1., a very relevant notion in the medical study context is selection bias (e.g., "A structural approach to selection bias" by Hernán et al., 2004). While I understand the fact that in this paper the domain generalization may not be limited to selection bias, the practical example presented (e.g., observational study vs. randomized trial) seems to be more aligned with the selection bias analysis. Further clarifications will be very helpful.

### 3. the limited significance of theoretical results

While it is nice to see a formalized connection between (certain types of) domain generalization to transportability theory, there is a worry about the significance of the theoretical results. For instance, Theorem 1 is directly applying Definition 3. Theorem 2 and Theorem 3 are, to a certain extent, direct outcomes of observational and interventional equivalences between different SCMs. The analysis on relative robustness is essentially just hierarchical relations between equivalence, observational equivalence, and interventional equivalence, which are established results in causal inference literature (e.g., "Elements of causal inference" by Peters et al., 2017).

### Some additional minor questions/comments:

- Is $Y$ limited to binary?

In introduction, what is the reason behind the "informative purpose" served by $0 < l < u < 1$?

- Before Section 2, "share an unobserved confounder", maybe "at least one" is more rigorous (just a minor thing, though).

- Typo: Section 2.1, "instead optimizing over" -> "instead of optimizing over"

- Typo: Section 2.1, "a chosen uncertainty set" -> "a chosen uncertain set"



**Summary Of The Paper:**

The paper presents the analysis of characterizing invariance learning with lowest worst-case loss as a special case of partial transportability tasks. The paper introduces the notion of "selection diagram" and considers domain generalization problems from the perspective of transportability analysis. Theoretical analysis on the bounds and some empirical analysis on the discrete variable case are also presented.

**Summary Of The Review:**

Overall, the paper makes an effort on connecting certain domain generalization problems to transportability analysis. Considering the lack of clarity in definition and configuration, there are worries about the correctness and the significance of the contribution (as detailed in "Strength and Weakness").

---

### Official Review · Reviewer_KuxZ · 2022-10-24

**Confidence:** 4
**Correctness:** 1
**Technical Novelty And Significance:** 3
**Empirical Novelty And Significance:** 3
**Recommendation:** 3

**Clarity, Quality, Novelty And Reproducibility:**

The paper is sensibly structured and generally clear. The core idea is original (though transportability and finding bounds already existed separately). There are ambiguous/ill-defined/wrong statements in some definitions and theorems.

**Strength And Weaknesses:**

The paper introduces an interesting new task, allowing progress to be made in cases where the usual notions of transportability are too restrictive.

Unfortunately, I encountered problems with several of the theoretical results in this paper.

* Definition 3: The condition that the subset is "the maximal subset" makes this ill-defined, and it is too simplistic for what you want to show in Theorem 1. The definition is ill-defined if there are multiple subsets that are each maximal w.r.t. inclusion, e.g. for the graph S -> A <-> B <-> Y. Then the Markov blanket of Y includes both A and B, but conditioning on both make Y and S dependent. Conditioning on either one gives a "maximal subset" that satisfies dependence. But for most distributions of the source domains, I expect that only one choice (usually Z={B}) will solve the worst-case optimization problem (2).

* Theorem 2: The phrase "the maximal subset" also appears here, making the statement of the theorem unclear if  there are multiple maximal subsets. (The proof of this theorem refers to two theorems in an earlier paper, but those make no mention of maximal subsets or Markov blankets.)

(The same phrase also appears in propositions 1 and 2.)

* Theorem 3: This is a suspect result - why would the lower bound have this special status? In the proof, I don't see why you can split the maximization over the terms when going from (16) to (17), or how the theorem's conclusion is drawn from (17). Also, (16,17) treats x as constant and then maximizes over it, while in (2), an expectation is taken over it inside the maximization. Can you explain these points?

Minor comments:

* (page 2) "is the statement" -> "if the statement"

* (page 4) "neurodegenrative" -> "neurodegenerative"

* final equation of section 3: $A \neq B \neq C$ does not imply $A \neq C$. I think the latter is what you want to say though. (I saw a similar construction elsewhere.)

**Summary Of The Paper:**

This paper introduces the partial transportability task. Unlike standard transportability, where a target quantity in a target domain must be point-identified, in partial transportability, it may be bounded instead. The theoretical development in the paper connects this new notion to existing theory. An algorithm for this task is introduced, based on Gibbs sampling.

**Summary Of The Review:**

Interesting idea, but I recommend rejection until the problems with the theoretical results can be resolved.

---

### Official Review · Reviewer_wQrN · 2022-10-24

**Confidence:** 2
**Correctness:** 3
**Technical Novelty And Significance:** 3
**Empirical Novelty And Significance:** 2
**Recommendation:** 6

**Clarity, Quality, Novelty And Reproducibility:**

I feel this work is addressing an important problem for domain generalization as existing approaches lack the measurement of such transportability. With some assumptions on causal graphs, the proposed method can quantify the transferability. Maybe one concern is that the authors put too many notations and definitions making this work a little bit hard to follow.

**Details Of Ethics Concerns:**

No ethic concern.

**Strength And Weaknesses:**

Pros:
- The paper is well-written and this work is, in general, of high quality.
- The authors introduced the proposed method in a comprehensive way with sufficient notations, definitions and theoretical claims.

Cons:
- It seems the authors considered discrete observations and finite domains. However, this setting is maybe too restrictive for many real-world problems.
- The proposed method originated from the robust optimization problem in Eq.(2), which optimize for the best predictor under the worst-case causal graph. While the authors claimed the bound leads to a distributional robust guarantee, they didn't explain the connection of robust g

Minor issues:
- Section 2, first paragraph: [What does the / mean?]
- Section 2, second paragraph, typo: factors such [as] physical activity levels


**Summary Of The Paper:**

Based on the existing formulation of the domain generalization problem, the authors proposed to find both the upper and lower bound performance on the empirical distribution of data from an unknow domain.
via MCMC method. The authors demonstrated that, following the notation of causal models, the gap between the upper and the lower bound characterizes the transferability towards a target domain.

**Summary Of The Review:**

It seems this work is well conducted. Under certain assumptions, the proposed method provides a new perspective to view the domain generalization problem.

---

### Official Review · Reviewer_pRFj · 2022-10-29

**Confidence:** 2
**Clarity, Quality, Novelty And Reproducibility:** The paper is well organized
**Correctness:** 3
**Technical Novelty And Significance:** 3
**Empirical Novelty And Significance:** 3
**Recommendation:** 6

**Strength And Weaknesses:**

Pros:

1. It’s novel to introduce partial transportability to the DG problem.

2. Instead of the point estimate, deriving a tight bound of the conditional expectation in the unseen domain is novel.

3. Illustrating invariance learning as a special case of partial transportability task is novel.


Cons:

1. A related work is necessary to demonstrate the background of the methods used in this paper, for example, transportability problem, etc.

2. Lacking datasets in experiment evaluations. The authors only evaluate the algorithm on one dataset.

3. The author should review [1] in detail and describe the differences because their proposed method has some overlaps with this paper.

[1] Zhang, Junzhe, Jin Tian, and Elias Bareinboim. "Partial counterfactual identification from observational and experimental data." International Conference on Machine Learning. PMLR, 2022.


**Summary Of The Paper:**

The paper formulates partial transportability to bound the query in the target domain in the domain generalization problem. The authors show that invariance learning is a special case of partial transportability tasks and show that invariant predictors and more general solutions to robust optimization problems. They evaluate the algorithm on a synthetic dataset.

**Summary Of The Review:**

This paper tackles the domain generalization problem with an under-studied causal method and models each domain as a different causal graph (SCM). Considering this paper researching an under-studied method in the DG field, I vote for a weak accept.

---

### Decision · Program_Chairs · 2023-01-20

**Decision:**

Reject

**Justification For Why Not Higher Score:**

Two reviewers are in favor of rejecting the paper.

**Justification For Why Not Lower Score:**

-

**Metareview: Summary, Strengths And Weaknesses:**

The paper studies the problem of domain generalization through the lens of transportability. Using the perspective of transportability theory, the authors show that invariance learning, and the settings in which invariant predictors are optimal in terms of worst-case losses, is a special case of a more general partial transportability task. The authors then propose solutions that highlight new contrasts with "invariance learning" methods.

The reviewers had a number of concerns, some of which were addressed via the authors' responses, but some other major ones remained. In particular, after the discussion with the reviewers, it was still apparent that the reviewers were still concerned about the correctness and the significance of the contribution. All the details are available in the reviews (I also took into account the messages that were sent by the authors).  One of the reviewers also (correctly) pointed out that the revised version is considerably above the allowed page limit.

All in all, all the reviewers felt that the paper provides a very interesting g perspective on domain generalization as well as a set of nice results. Once the concerns of the reviewers are addressed, the paper will be a very interesting contribution to the area of domain generalization.